# Evaluation of the Functional Status of the Posture Control System in Children with Detected Disorders in Body Posture

**DOI:** 10.3390/ijerph192114529

**Published:** 2022-11-05

**Authors:** Anna Permoda-Białozorczyk, Marzena Olszewska-Karaban, Andrzej Permoda, Jolanta Zajt, Marek Wiecheć, Arkadiusz Żurawski

**Affiliations:** 1Biomed Rehabilitation Center, ul. Dębinki 7d, 80-294 Gdańsk, Poland; 2Division of Rehabilitation Medicine, Faculty of Health Sciences, Medical University of Gdańsk, al. Zwycięstwa 30, 80-219 Gdańsk, Poland; 3Department of Health and Natural Science, Gdańsk University of Physical Education and Sport, Kazimierza Górskiego 1, 80-336 Gdańsk, Poland; 4Centrum Rehabilitacji MARKMED, ul. Iłżecka 31a, 27-400 Ostrowiec Świętokrzyski, Poland; 5Institute of Health Science, Collegium Medicum, Jan Kochanowski University, ul. Żeromskiego 5, 25-369 Kielce, Poland

**Keywords:** body posture defects, proprioceptive disorders, postural instability, screening tests

## Abstract

Screening tests for body posture defects and abnormalities conducted over the past several decades have revealed a significant and constantly increasing problem of health risks in children. A sedentary lifestyle, which is considered to be the primary cause, can result in proprioceptive disorders leading to postural instability. The aim of the study was to find the correlation between the level of proprioceptive control and the number of postural disorders. The study involved a representative group of 1090 children aged 8–10 years, from randomly selected primary schools. Subjects who scored from 1 to 5 points in a prior postural screening test were qualified for the examination of the postural control system. The examination of the postural control system was carried out using an electronic station enabling assessment of postural stability and proprioception. A single leg stance test showed that the number of postural disorders does not significantly impact proprioceptive control. Proprioceptive control was found to significantly increase with the age of the children, and girls presented significantly better proprioceptive control in relation to the boys in each age group.

## 1. Introduction

The process of posturogenesis is accompanied by a continuous, conditioned, and encoded development of postural reflex and locomotion. A normal process of body posture development depends on numerous exogenous and endogenous factors [1]. Decreased physical activity is seen as the most frequent cause of postural defects, since less and less proprioceptive information reaches the nervous system, which may disrupt the natural posturogenesis process and lead to postural instability [2]. As a result, muscle tonus and alignment of individual body segments are disturbed leading to an abnormal, fixed body shape—a postural defect.

Screening tests designed to detect postural defects and abnormalities, conducted over the past several decades, have shown there is a significant and constantly growing problem of health risks in children [3,4,5,6]. Epidemiological data on the prevalence of postural defects in school-age populations are greatly varied. According to Kowal, school children with postural defects account for approximately 50% of the populations studied [7,8]. Kaczmarek and Raczkowski report approximately 70% prevalence of postural defects in the city of Łódź [9]. In the region of Zielona Góra, postural disorders were found in 77% of the subjects, with most abnormalities occurring in 6 and 7-year-old children. Research carried out in the Świętokrzyskie Region showed that the problem of postural defects affected as many as 80% of children [10]. Unfortunately, it is not only the high prevalence of postural abnormalities that is worrying. Numerous studies suggest that the number of children with musculoskeletal defects will increase each year [8,10,11,12,13]. Currently, a decreasing level of physical fitness and growing rates when overweight and obese can also be observed in populations of preschool and school-age children [10,11,14,15,16]. It results primarily from the sedentary lifestyle, associated with low physical activity and a tendency to spend free time in front of a computer or TV [10,13]. The general results reported in 2008 by the National Institute of Public Health indicate that the physical activity of Poles is too low, and the activity of children and adolescents is decreasing in older age groups [17]. Low physical activity is one of the main causes of disorders and postural defects in children and adolescents [18,19,20,21]. Despite the advancements in physiotherapeutic methods enabling treatment of postural defects, and the gradually implemented preventive and treatment programs aimed at reducing the problem of postural abnormalities, the relevant rates are not decreasing; on the contrary, they are constantly growing. In addition to the emerging number of similar prevention and treatment programs, it is necessary to look closely at the mechanisms that may affect the incidence of the above-mentioned problems.

## 2. Materials and Methods

### 2.1. Subjects

A group of 1090 individuals, 564 girls and 526 boys aged 8–10 years from selected primary schools of the city and municipality of Zielona Góra, were enrolled in the study. The examinations were carried out between 2009 and 2010 as a part of a program entitled “Prosto do zdrowia”, focusing on postural defects prevention, as well as diagnostics, and aimed toward prophylaxis and therapy of postural defects, and at improving the overall level of health and physical fitness in children and adolescents. The specific objective of the program was to also shift the emphasis from treatment to prevention in accordance with current guidelines for health care development. The characteristics of the study group are presented in Table 1.

Individuals were assessed for the presence of body posture defects. The individuals who were diagnosed with at least one postural defect were qualified for the specialist examination performed using a Delos Postural System Professional 4.0 device (Delos s.r.l., Turin, Italy). The tests were carried out by a specialist in physiotherapy with a doctoral degree in medical sciences.

Points were awarded depending on the occurrence of:
Disturbed alignment of the spine in the frontal plane;Disturbed alignment of the spine in the sagittal plane;Scoliosis (positive result in the Adam’s test);Knee malalignment (valgus or varus);Foot malalignment (valgus, varus or flatfoot).

One posture defect scored one point. Thus, the respondents could obtain a result from 0 (no posture defect) to 5 points (presence of all 5 assessed posture defects).

The inclusion criteria were obtaining from 1 to 5 points in the body posture examination by visual assessment and Adam’s test.

The exclusion criteria were as follows:-physical disability;-mental disability;-developmental anomalies of the feet;-true shortening of a lower limb;-sensory integration disorders;-neurological conditions;-nystagmus;-attention deficit disorder;-feeling unwell on the day of the examination.

Parents/legal guardians of the subjects expressed their written consent for the examination.

The use of the research for this paper was approved by the Independent Bioethics Committee of the Medical University of Gdańsk, dated 19 September 2011, resolution number: NKEBN/359/2011, and accepted in writing by the coordinator of “Prosto do zdrowia” program.

### 2.2. Procedures

The test was performed under similar lighting and time conditions by the same person.

#### 2.2.1. Instruments

The stability test was performed by means of electronic postural proprioceptive station (DPPS). The station, connected to a personal computer with specific software (DPPS 4.0), consisted of an electronic postural reader, an infrared sensor bar, and a display (Figure 1). In case of falling, the subject could touch the bar placed in front of them to regain vertical control rapidly. The bar was equipped with an infrared sensor which registered each contact when support was needed, and sent information to the computer software (DPA) [4,5,6]. The electronic postural reader (Delos Vertical Controller-DVC), applied to the sternum, measured the trunk inclination in the frontal (x) and sagittal plane (y) by means of a 2-dimensional accelerometer unit. The rotation range of this sensor, when mounted on the sternum, was between 0 and 15 cm, and never exceeded 30 cm [3,4]. The static platform was a wooden base that allowed the provision of uniform conditions during tests and static training.

#### 2.2.2. Algorithms

The data from the postural reader were a stream of acceleration samples taken by converting the sensor outputs into the digital domain at a rate of 100 Hz. These raw data were initially averaged with a 4-tap sliding window; therefore, the 3 dB bandwidth was narrowed to approximately 11 Hz. Scaling with the calibration data of the instruments was then performed to convert the raw data into angles. The equations involved in the management of data from the postural reader have been described in a previous article [3].

#### 2.2.3. Indicator of Proprioceptive Control

The stability index with closed eyes, SI CE (%), was used to assess proprioceptive control. This index allows researchers to grade all tests and to compare individuals with high stability scores obtained during the test (very narrow postural cone and complete autonomy) and those with a lower level of stability (very low autonomy of the system) [3].

#### 2.2.4. Single Stance Stability Assessment

Static Riva Test was used to assess the functional status of the postural control system. It was performed on an electronic station for postural stability and proprioception tests, Delos Postural System Professional 4.0.

Static Riva Test was performed using Delos Vertical Controller (DVC), which was attached to the subject’s sternum, and Delos Postural Assistant (DPA), which the subjects were asked to use as hand support in case of imbalance. The DPA was adjusted to each person’s height to provide adequate support. 

The device was connected to a computer with specialized software (PSM 4.0) designed to receive and analyze information from the specific parts of the device.

Static Riva Test consisted of 20 s trial of one-leg stance on the left leg with eyes open, then 20 s on the right leg with eyes open, followed by the same amount of time on the left leg with eyes closed, and on the right leg with eyes closed. Every 20 s trial was followed by a 15 s interval.

### 2.3. Statistical Analysis

All calculations were performed using Microsoft Excel version 2007 and Statistica version 8.5 statistical package. In the statistical description of the quantitative data, classic position measurements were used, such as arithmetic mean, median, and mode, as well as the standard deviation and the definition of minima and maxima as general measures of variation.

When comparing two groups for quantitative and stepwise data, the Mann-Whitney U test (comparison of two groups) was used, and the Wilcoxon test (comparison of two measurements with the same people). In the comparative analysis of a larger number of nonparametric data groups, the Kruskal-Wallis test was used together with the post hoc multiple test (Dunn test) as a non-parametric equivalent of ANOVA, the parametrically evaluated method of analysis of variance (ANOVA) using the Fischer test, and the post hoc tests Scheffegi and RIR Tukey. In order to assess the strength, direction, and significance of linking variables, Pearson’s linear correlation coefficient was used. In all statistical tests, *p* < 0.05 was assumed as the level of statistical significance of differences.

Statistical analysis was carried out in cooperation with the School of Medical Informatics and Biostatics at the Medical Faculty of the Medical University of Gdańsk.

## 3. Results 

The influence of the number of postural disorders on the level of proprioceptive control index in the children aged 8 years.

Table 2 presents the mean values, median, and standard deviation of the index of stability with closed eyes in children aged 8 years with a different number of postural disorders.

The Kruskal-Wallis test did not show any statistically significant differences between the level of SI CE and the number of postural disorders in children aged 8 (*p* = 1) (Figure 2). 

The Mann-Whitney U test showed that the level of stability index of the girls was significantly higher than the level of stability index of the boys (Figure 3).

The influence of the number of postural disorders on the level of proprioceptive control in the children aged 9 years.

Table 3 presents the mean values, median, and standard deviation of the stability index with closed eyes in children aged 9 years with a different number of postural disorders.

The Kruskal-Wallis test did not show any statistically significant differences between the level of the stability index closed eyes and the number of postural disorders in the children aged 9 (*p* = 0.4) (Figure 4). 

The differences between the boys and the girls aged 9 years were found to be statistically significant (*p* < 0.0001). The level of the stability index for the girls was significantly higher than the level of the stability index for the boys (Figure 5).

The influence of the number of postural disorders on the level of proprioceptive control indicator in the children aged 10 years.

Table 4 presents the mean values, median, and standard deviation of the stability index with closed eyes in the children aged 10 years with a different number of postural disorders.

The Kruskal-Wallis test did not show any statistically significant differences between the level of the stability index with closed eyes and the number of postural disorders in the children aged 10 years (*p* = 0.5) (Figure 6). 

The mean level of the stability index in the group of girls and boys differed in a statistically significant way (*p* < 0.0001). The Mann-Whitney U test showed that the level of the stability index for the girls was significantly higher than the level of the stability index for the boys representing this age group (Figure 7).

The influence of age on the level of proprioceptive control index in the children aged 8–10 years with postural disorders.

Table 5 shows the mean values, median, and standard deviation of the stability index with closed eyes in all the examined age groups.

The Kruskal-Wallis test showed statistically significant differences between the level of the stability index with closed eyes and the age of the subjects (*p* < 0.0001). A multiple comparison test was performed to identify the age group with the highest differences. The test showed that all the age groups differed from one another in a statistically significant manner, and the level of the stability index with closed eyes visibly increased with age. The relationships are shown in detail in Figure 8.

## 4. Discussion

According to Riva, decreased physical activity results in reduced proprioceptive information supplied, which consequently leads to poorer neuromuscular control causing functional instability of the joints and decreased balance capacity, which may eventually lead to postural instability [2,5]. Dynamic position control tests examining a one leg stance, carried out using a Delos Postural System Professional 4.0 device, in a population of children aged 9–13, indicated that children who lead a sedentary lifestyle obtained significantly poorer results compared to children under taking various sports [2]. During 30-s trials, 96% of the subjects involved in exercise obtained complete autonomy after 3 s, while 43% of the children who led a sedentary lifestyle were not able to complete the test. Posturographic tests are another objective method used for examining the stability of the body. The assessment of the state of equilibrium is made in a standing position, based on the analysis of the obtained graphical records for the displacement of the center of gravity of the body [22]. These static and dynamic tests are performed using a posturographic platform supported by a computer system [22]. A study carried out by Brzęk, Nowotna-Czupryna, and Famuła, with trials carried out on a platform tilted in different directions, showed a poorer ability to assess the vertical and horizontal line in the children diagnosed with scoliosis compared to the healthy children; the boys also obtained poorer results [23]. The authors also showed that the vertical and horizontal placement of points in space is varied in school-age children [23]. Likewise, Ostałowska et al. also emphasized that the ability to restore body balance following a disturbance is visibly poorer in children and adolescents with scoliosis, and the magnitude and character of this type of reflex response is affected by the degree of spinal deformation [24]. The authors reported that the greatest differences in the ability to maintain a stable posture and to restore equilibrium were observed in children with scoliosis exceeding 40 degrees. These differences were minor in subjects with smaller spinal curvatures. No significant differences were found in children with normal posture and minor spinal curvatures [24]. Other studies carried out by Simoneau et al. suggested a different cognitive integration of signals originating from the vestibular system in patients with diagnosed idiopathic scoliosis compared to healthy persons [25].

The current study showed significant age-related differences in the scores identified during one leg stance trials. Posture stability level constantly increased from 8 to 10 years of age. This is consistent with results reported by Morioka, who believed that the ability to maintain a standing position on one leg with eyes open rapidly improves from late pre-school to early school age, and then slows down during later school age. Statistically significant differences were also observed between girls and boys. The results indicate that girls have a better ability to maintain a stable posture in a one leg stance, compared to boys in every group from 8 to 10 years of age.

On the other hand, the present study has shown that the number of postural defects does not significantly impair posture stability in any of the age groups from 8 to 10 years. In some age groups it is possible to only observe a tendency towards a decrease in the stability index associated with a larger number of postural defects. In the 8-year-old group, the downward trend was observed only if there were more than two defects in body posture. In the group of children aged 9 years, a similar tendency was observed in cases with up to three postural defects. In 10-year-olds, poorer results were observed only in the group of children with three defects in comparison to the children with two defects. The lack of a downward trend in the stability index in the groups with more than three disorders in body posture may be linked to the small size of the groups with four or five defects, which is reflected by a relatively large range of standard error. It should also be considered that postural defects were not rated by the authors for the quality of the disorder, but only in terms of their presence. Further research should also consider the degree of the disorder, as it is possible that the stability index will deteriorate with increasing defects. Studies by Herdea et al. have shown that some cases of idiopathic scoliosis occur and progress due to insufficient levels of vitamin D, calcium, and melatonin [26]. Therefore, future research on the causes of abnormalities in body posture could also be extended to include a greater number of factors.

## 5. Conclusions

Proprioceptive control significantly increases with age in children with diagnosed postural defects.Girls aged 8 to 10 years had statistically significantly better results in proprioceptive control compared to boys, in all age groups.The number of postural defects does not significantly impair posture stability in any of the age groups from 8 to 10 years.The number of postural defects has no significant effect on proprioceptive control.

## Figures and Tables

**Figure 1 ijerph-19-14529-f001:**
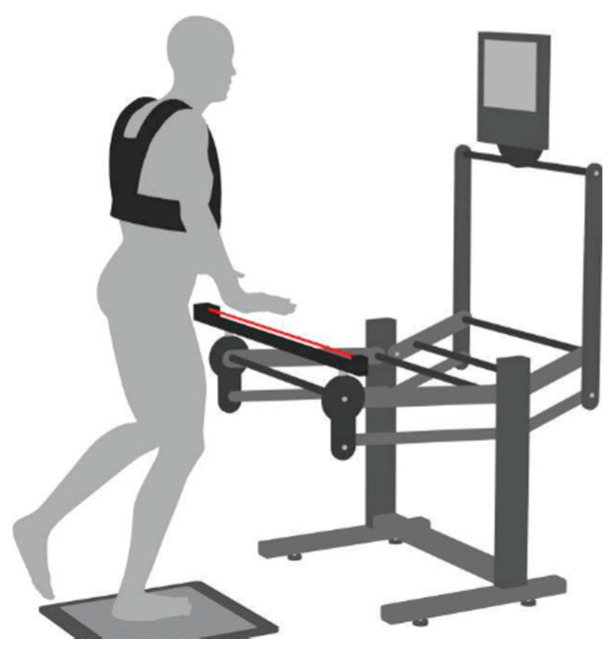
The postural proprioceptive station. The red line represents an infrared ray of the sensorised bar (DPA). Vest to support the “postural reader”, a two-dimensional accelerometer unit, in sternal position (DVC), and static platform [3].

**Figure 2 ijerph-19-14529-f002:**
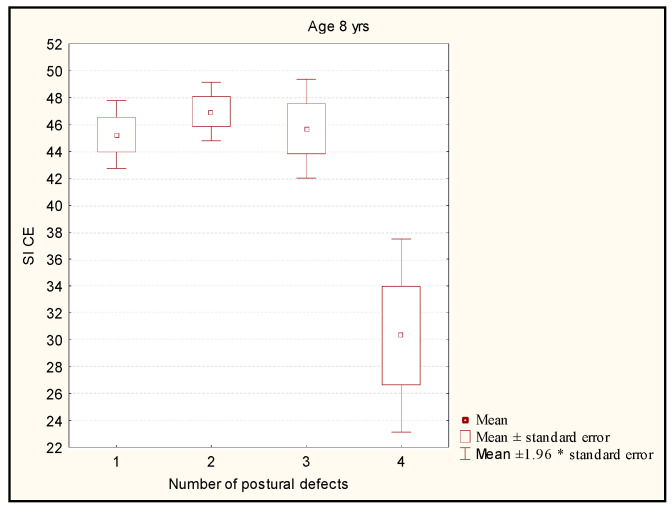
Comparison of the stability index level in the groups with different numbers of postural disorders in the children aged 8. Legend: SI CE—stability index closed eyes; *—multiplication.

**Figure 3 ijerph-19-14529-f003:**
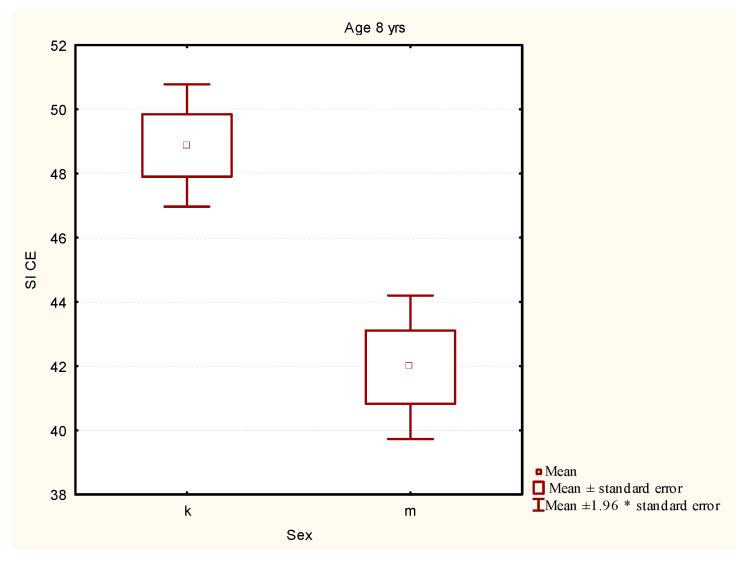
Comparison of the stability index level in the girls and the boys aged 8 years. Legend: SI CE—stability index closed eyes (mean value of stability index with closed eyes on left and right leg); k—girls, m—boys; *—multiplication.

**Figure 4 ijerph-19-14529-f004:**
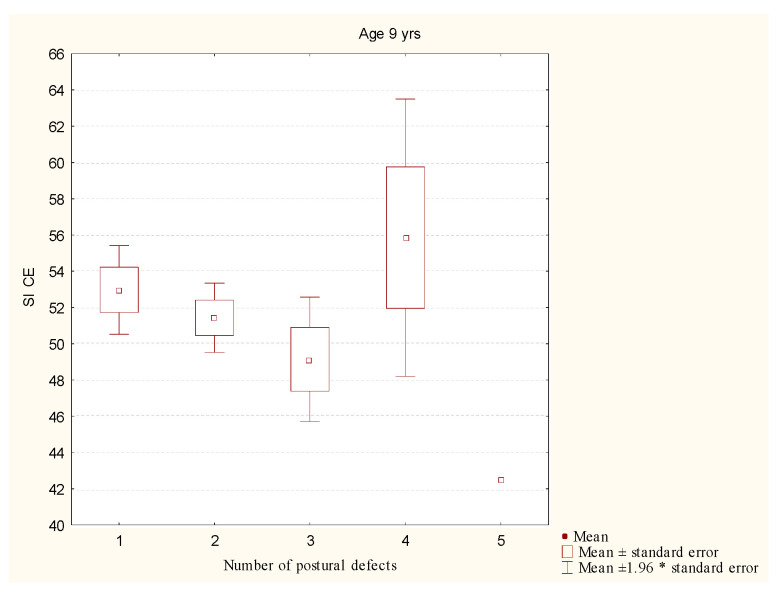
Comparison of the level of the stability index in the groups with different numbers of postural disorders in the children aged 9. Legend: SI CE—stability index closed eyes; *—multiplication.

**Figure 5 ijerph-19-14529-f005:**
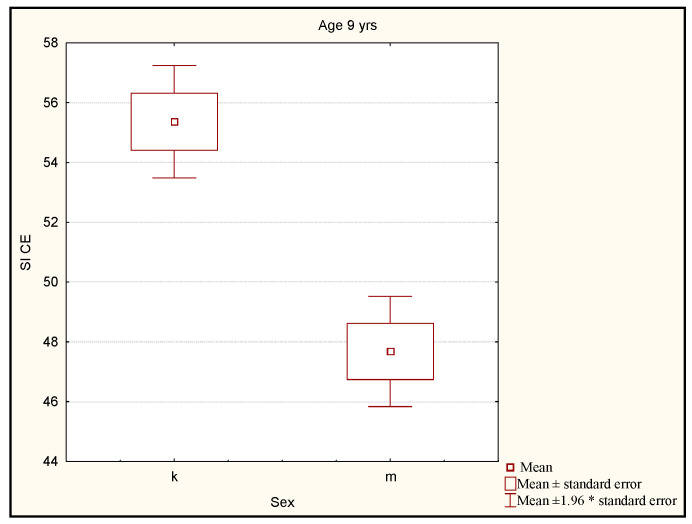
Comparison of the stability index level for the girls and the boys in the group of 9-year old subjects. Legend: SI CE—stability index closed eyes (mean value of stability index with closed eyes on left and right leg); k—girls; m—boys; *—multiplication.

**Figure 6 ijerph-19-14529-f006:**
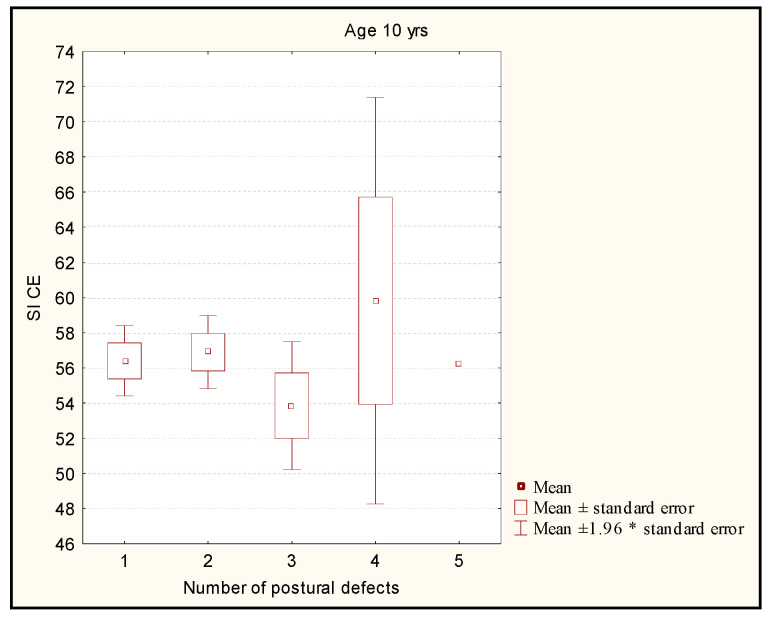
Comparison of the level of the stability index in the groups with different numbers of postural disorders in the children aged 10 years. Legend: SI CE—stability index closed eyes; *—multiplication.

**Figure 7 ijerph-19-14529-f007:**
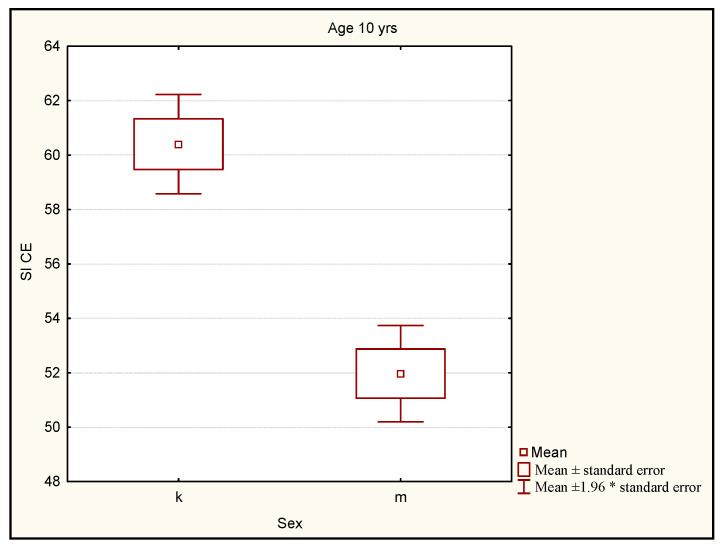
Comparison of the stability index level for girls and boys in the group of 10-year-old subjects. Legend: SI CE—stability index closed eyes (mean value of stability index with closed eyes onleft and right leg); k—girls; m—boys; *—multiplication.

**Figure 8 ijerph-19-14529-f008:**
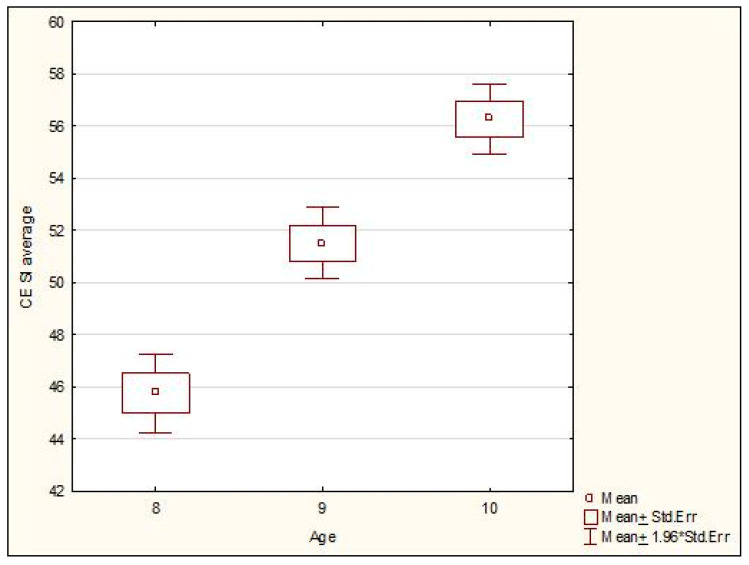
Comparison of the level of the stability index closed eyes in all the examined age groups. Legend: CE SI—stability index closed eyes (mean value of stability index with closed eyes on left and right leg); Std. Err—standard error; *—multiplication.

**Table 1 ijerph-19-14529-t001:** The total number and percentage of the subjects with respect to sex and age.

Age of the Subjects	G + B		G		B	
	N	% *	*n*	% **	N	% ***
8 yrs	333	30.6	183	32.5	150	28.5
9 yrs	393	36	196	34.7	197	37.5
10 yrs	364	33.4	185	32.8	179	34
Total	1090	100	564	100	526	100

Legend: N—total number of subjects in the age group; *n*—number of girls (boys) in the age group; G + B—the number of girls and boys examined in the age group; G—girls; B—boys: * the percentage of all individuals in the age group; ** percentage of all girls in the age group; *** percentage of all boys in the age group.

**Table 2 ijerph-19-14529-t002:** The level of the index of stability with closed eyes in children aged 8 years with a certain number of postural disorders.

Age 8 yrs			
SI CE			
Number of Disorders	N	Mean	SD
1 disorder	122	45.25	14.21
2 disorders	144	46.96	13.28
3 disorders	60	45.69	14.47
4 disorders	7	30.31	9.69
5 disorders	no cases identified		

Legend: N—number of subjects in the age group with the specific number of postural disorders; SI CE—stability index closed eyes; SD—standard deviation.

**Table 3 ijerph-19-14529-t003:** The level of the stability index with closed eyes in children aged 9 years with a certain number of postural disorders.

Age 9 yrs			
SI CE			
Number of Disorders	N	Mean	SD
1 disorder	117	52.97	13.51
2 disorders	196	51.43	13.64
3 disorders	73	49.14	14.97
4 disorders	6	55.85	9.55
5 disorders	1	42.52	0

Legend: N—number of subjects in the age group with the specific number of postural disorders; SI CE—stability index closed eyes; SD—standard deviation.

**Table 4 ijerph-19-14529-t004:** The level of the stability index with closed eyes in the children aged 10 years with a certain number of postural disorders.

Age 10 yrs			
SI CE			
Number of Disorders	N	Mean	SD
1 disorder	140	23.03	9.66
2 disorders	154	23.18	11.33
3 disorders	61	25.06	10.16
4 disorders	8	21.40	17.69
5 disorders	1	24.07	0

Legend: N—number of subjects in the age group with a specific number of postural disorders; SI CE—stability index closed eyes; SD—standard deviation.

**Table 5 ijerph-19-14529-t005:** The level of the stability index with closed eyes in all the examined age groups.

SI CE			
Age	N	Mean	SD
8 yrs	333	45.76	13.94
9 yrs	393	51.51	13.82
10 yrs	364	56.25	13.06

Legend: N—total number of subjects in a given age group; SI CE—stability index closed eyes (mean value of stability index with closed eyes left and right leg); SD—standard deviation.

## Data Availability

The data presented in this study are available on request from the corresponding author. The data are not publicly available due to privacy.

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
