# Peer review of "Evaluation of the Functional Status of the Posture Control System in Children with Detected Disorders in Body Posture"

_ijerph, 2022, doi:10.3390/ijerph192114529_

Round 1

Reviewer 1 Report

Dear author, 

This work may not add something new to the literature, but it certainly builds on the belief that physical activity and postural balance are related. It is a well-prepared work and it might be a candidate for publication in this journal.

The introduction is brief and precise. Among Subjects, you did not mention the inclusion criteria, only the exclusion criteria. Methods are succinct and a bit synthetic. 

It would be interesting to mention in the Discussion other clinical measurement tools for proprioception besides “SI CE” to make room for further studies in the field of interest, and other correlations found in the literature to complete the information from this specific area.

Table 6 needs to be completed or removed. 

You concluded that the number of postural defects is irrelevant, but you stated that the severity of the defect is important (as in row 326). If supported by statistical analysis, it should be stated as another conclusion that could be very important for the field of Orthopaedics.

We found that calcium and vitamin D homeostasis is also important in the etiology and treatment of deformities and we encourage you to read the article and use it as a reference: Controlling the Progression of Curvature in Children and Adolescent Idiopathic Scoliosis Following the Administration of Melatonin, Calcium, and Vitamin D, published in Children, DOI 10.3390/children9050758.

Author Response

Thank you very much for all your comments. We tried to correct all the submitted comments. I will refer to them in turn below.

The introduction is brief and precise. Among Subjects, you did not mention the inclusion criteria, only the exclusion criteria. Methods are succinct and a bit synthetic. 

Answer: This part has been completed. The inclusion criteria indicated that children with at least one posture deficit were enrolled in the study.

We did not excessively expand the part describing the method, because the research is based on the use of specific devices. Only the mechanism of their action is described.

 It would be interesting to mention in the Discussion other clinical measurement tools for proprioception besides “SI CE” to make room for further studies in the field of interest, and other correlations found in the literature to complete the information from this specific area.

Answer: Thank you for this suggestion. The discussion was supplemented with two other methods of assessing this phenomenon and the results obtained with their use.

Table 6 needs to be completed or removed. 

Answer: After talking to the co-authors, we decided to delete this table. thank you for paying attention to it.

You concluded that the number of postural defects is irrelevant, but you stated that the severity of the defect is important (as in row 326). If supported by statistical analysis, it should be stated as another conclusion that could be very important for the field of Orthopaedics.

Answer: Thank you for this suggestion. Another request added: "3. The number of postural defects does not significantly impair posture stability in any of the age groups from 8 to 10 years." The need to assess the impact of the size of postural defects on the described phenomenon was included in the part concerning the limitations of the study.

We found that calcium and vitamin D homeostasis is also important in the etiology and treatment of deformities and we encourage you to read the article and use it as a reference: Controlling the Progression of Curvature in Children and Adolescent Idiopathic Scoliosis Following the Administration of Melatonin, Calcium, and Vitamin D, published in Children, DOI 10.3390/children9050758.

Answer: Thank you for your attention. We have attached the above information and the suggested reference to the discussion.

Reviewer 2 Report

Thank you for the opportunity to review this paper. It is a well written paper with an interesting topic. 

Minor suggestions:

- Line 83-84 are not written in clear English. I suggest a re-write.

- It is not made adequately clear about how participants scored points. I think this needs to be clarified for ease of readability

- Is there a reason for presenting both mean and median?

Author Response

Thank you very much for all your comments. We tried to correct all the submitted comments. I will refer to them in turn below.

Line 83-84 are not written in clear English. I suggest a re-write.

Answer: This paragraph has been rewritten to make it clearer. It now reads: „The individuals were assessed for the presence of body posture defects. The individuals who diagnosed with at least one postural defects were qualified for the specialist examination performed using Delos Postural System Professional 4.0 device. The tests were carried out by a specialist in physiotherapy with a doctoral degree in medical sciences.  “

It is not made adequately clear about how participants scored points. I think this needs to be clarified for ease of readability.

Answer: Thank you for this suggestion. This issue has been supplemented: „One posture defect scored one point. Thus, the respondents could obtain a result from 0 (take any posture defect) to 5 points (presence of all 5 assessed posture defects).”

Is there a reason for presenting both mean and median?

Answer: Thank you for drawing attention to this issue. The mean in the tables has been included for easier observation between groups. The median was included due to the use of non-parametric tests in some of the calculations. After discussing with the co-authors, we decided to leave only the average in the tables. Now the tables are much clearer.

Round 2

Reviewer 1 Report

The paper is ready to be publish